# Within-Household Transmission and Bacterial Diversity of *Staphylococcus pseudintermedius*

**DOI:** 10.3390/pathogens11080850

**Published:** 2022-07-28

**Authors:** Alice Wegener, Birgitta Duim, Linda van der Graaf-van Bloois, Aldert L. Zomer, Caroline E. Visser, Mirlin Spaninks, Arjen J. Timmerman, Jaap A. Wagenaar, Els M. Broens

**Affiliations:** 1Department of Infectious Diseases and Immunology, Faculty of Veterinary Medicine, Utrecht University, 3584 CL Utrecht, The Netherlands; a.c.h.wegener@uu.nl (A.W.); l.vandergraaf@uu.nl (L.v.d.G.-v.B.); a.l.zomer@uu.nl (A.L.Z.); a.j.timmerman@uu.nl (A.J.T.); j.wagenaar@uu.nl (J.A.W.); e.m.broens@uu.nl (E.M.B.); 2Department of Medical Microbiology & Infection Prevention, Amsterdam UMC Location AMC, Amsterdam, 1105 AZ Amsterdam, The Netherlands; c.e.visser@amsterdamumc.nl; 3Department Population Health Sciences, Faculty of Veterinary Medicine, Utrecht University, 3584 CL Utrecht, The Netherlands; m.p.spaninks@uu.nl; 4Wageningen Bioveterinary Research, 8221 RA Lelystad, The Netherlands

**Keywords:** *S. pseudintermedius*, transmission, One health, whole genome sequencing, zoonotic, bacterial diversity

## Abstract

*Staphylococcus pseudintermedius* can be transmitted between dogs and their owners and can cause opportunistic infections in humans. Whole genome sequencing was applied to identify the relatedness between isolates from human infections and isolates from dogs in the same households. Genome SNP diversity and distribution of plasmids and antimicrobial resistance genes identified related and unrelated isolates in both households. Our study shows that within-host bacterial diversity is present in *S. pseudintermedius*, demonstrating that multiple isolates from each host should preferably be sequenced to study transmission dynamics.

## 1. Introduction

*Staphylococcus pseudintermedius* is both a commensal and opportunistic pathogen in dogs. Infections in humans are occasionally found; however, in humans, *S. pseudintermedius* might be underdiagnosed as it can be misidentified as *Staphylococcus aureus* or *Staphylococcus intermedius* [1,2]. Human infections with *S. pseudintermedius* are generally considered to be of zoonotic origin [3], although in exceptional cases no dog contact is reported [4]. Dog-to-human transmission of *S. pseudintermedius* has been reported, in which isolates from dogs and their owners were indistinguishable based on multi-locus sequence typing and pulsed field gel electrophoresis [4,5]. Nevertheless, carriage rates of *S. pseudintermedius* in humans remain very low compared to the carriage rates of dogs, even in dog owning households [6]. Longitudinal studies on methicillin-resistant *S. pseudintermedius* (MRSP) showed that dogs carried MRSP for prolonged periods of time (several months), whereas carriage in humans was rare and short-term. Human carriage is therefore considered to be contamination instead of colonization, though opportunistic infections in humans can occur [5,7]. In longitudinal studies, MRSP was found in the environment and in other dogs in the household [7,8]. Generally, isolates within one household belong to the same sequence type (ST), although occasionally different STs can be found in the same household [5]. Most studies on dog-to-human transmission of *S. pseudintermedius* include only a single isolate from each host. This approach might lead to misinterpretations when within-host bacterial diversity exists. We used whole genome sequencing of multiple isolates from dogs to investigate within-household transmission and bacterial diversity of *S. pseudintermedius* in two unrelated human infections caused by *S. pseudintermedius* and the dogs in these households.

## 2. Results

### 2.1. Household 1

Patient 1 was a 64-year-old woman with a wound infection on her foot in June 2016. One dog, suffering from a chronic skin condition, was present in the household. S. pseudintermedius was isolated from three sampling sites and multiple isolates were selected for genome analysis (n = 5 from each site) based on morphological colony differences. This provided insight into the number of single nucleotide polymorphisms (SNP) in isolates from this dog. Dog isolates belonged to two clades that differentiated by 6913 core-genome SNPs. One clade consisted of six dog isolates (obtained from perineum and axillary) that displayed a very low level of diversity (differing by up to 5 SNPs) and belonged to ST387 (Figure 1). All six dog isolates carried the *blaZ* resistance gene and no plasmid was detected.

In the other clade, the human isolate and nine of the dog isolates, obtained from the skin and axillary, differentiated between 0 and 7 core-genome SNPs and belonged to ST1337 (Figure 1). All isolates carried the resistance gene *tet(M)*, and all but one (16S06095-5) isolate carried the *blaZ* gene. The human isolate carried the *blaZ* and *tet(M)* resistance genes, no plasmid sequences, and differed by 7 SNPs from a dog isolate from the same household that also carried these genes and no plasmid sequences (Table 1).

### 2.2. Household 2

Patient 2 was a 63-year-old woman with an infected skin ulcer in July 2017. Three dogs were present in the household. No clinical conditions were reported for the dogs. All dogs were found to be positive for *S. pseudintermedius*, but not for all sites. Selection of morphologically different colonies resulted in one isolate from the skin of dog 1, one isolate from the perineum of dog 2, and 5 isolates from the skin (n = 3), the perineum (n = 1), and axillary (n = 1) of dog 3. 

The human isolate, and all isolates from dog 1 and dog 3, belonged to ST241. The ST241 isolates from dog 1 and dog 3 differed by between 0 and 9 SNPs, whereas the human isolate showed 87 SNPs differed from its closest related canine isolate (dog 1). In the MS-tree, the SNPs were filtered for recombination and the 87 SNPs between dog and human isolates were not clustered in one location on the genome, indicating that these SNPs were not the result of a single recombination event. Isolate 17S01590-2 of dog 2 displayed 7835 SNPs compared to its closest relative, belonged to ST940, and was considered genetically unrelated to other isolates (Figure 1). All isolates of household 2 carried the p222 plasmid (coverage 97%, identity 99%) [9] and other predicted plasmid sequences. The BLASTn analysis of these contigs identified sequence homology with the PRE-25-like element [10], carrying *sat4*; *ant(6)-Ia*; *aph(3’)-III*; *cat(pC221)*; and *erm(B)* resistance genes (coverage 67.9%, identity 99.9%) in all these isolates. A 2.7 kb plasmid sequence in all ST241 dog isolates belonged to the rep21 gene plasmid family (Table 1).

## 3. Discussion

Whole genome sequencing of *S. pseudintermedius* isolates from two unrelated human infections showed very low SNP diversity with canine isolates of colonized dogs in both households. The isolates retrieved from the human infections were considered genetically related to the isolates of the dogs. This is in accordance with longitudinal studies on MRSP showing that generally similar or indistinguishable *S. pseudintermedius* isolates can be present in humans, dogs, and environmental samples within the same household [5,7].

This study analyzed multiple dog isolates in one household, as it is known that inferring transmission by sequencing single colonies can be hindered by within-host bacterial diversity [11,12]. The SNP diversity in the genomes between several of the studied dog isolates in household 1 was very low, most likely reflecting the diversity that occurs during colonization. However, the genomes with higher SNP diversity (6913 and 7835) indicated that dogs were colonized with genetically unrelated isolates. This highlights the need for sequencing multiple isolates from dogs to investigate household transmission. SNP diversity correlated with assigned MLST sequence types as isolates from the same ST generally carried less than 10 SNP differences, whereas isolates with different STs differentiated by either 6913 or 7835 SNPs. Sequence type and SNP differences between MSSP isolates of different body sites were also observed, with dogs being positive for either one or multiple body sites, with different frequencies for each site [13]. This study also showed that isolates presenting morphological differences can be very closely related.

Mobile genetic elements were identified in all isolates from household 2: the p222 [9] and the PRE25-like elements. The presence of these elements in CC241 isolates, and the presence of this clonal complex in human isolates, has been previously reported [14]. The plasmid present in dog isolates in household 2 was absent in the ST241 human isolate and shows that gain or loss of a plasmid occurred among highly genetically related isolates. This is in line with observed gene loss or acquisition events in *S. aureus,* which is involved in the host jump of CC398 from livestock to human, and there are other examples of gene acquisitions in *S. aureus* that have facilitated adaptations to other animal species [15,16]. The mobile elements in *S. pseudintermedius* carrying multiple resistance genes and potential virulence genes are important epidemiological markers to monitor, as they can act as a reservoir for transmission to humans [14]. Nevertheless, the genome comparison showed that ST241 isolates from dogs in household 2 were more closely related to each other (<10 SNPs) than to the human isolate (87 SNPs). The higher SNP diversity might suggest that evolution occurred over the course of infection, but as the patient was only sampled once this could not be confirmed. Furthermore, as the dogs in this study were sampled within the same month the patients were hospitalized, it is impossible to infer the direction and timing of transmission. It would be interesting to have multiple samples from the human to see if the diversity observed in the dog is also present in human hosts. Larger studies using whole genome sequencing combined with epidemiological data would be of interest to determine if SNP differences between related isolates are common and can indicate the direction of transmission.

## 4. Materials and Methods

### 4.1. Bacterial Isolates 

*S. pseudintermedius* isolates from two human patients were obtained from the Amsterdam UMC location AMC in the Netherlands. Both patients were dog owners and gave their consent for samples from their dog(s) to be taken. Dogs (one in household 1 and three in household 2) were sampled by the owner at three body sites (skin, perineum, and axillary) within a month of confirmed infection of the owner. Samples were inoculated on sheep blood agar (bioTRADING, Mijdrecht, The Netherlands) and, after overnight incubation at 37 °C, presumptive colonies were identified as *S. pseudintermedius* by Maldi-TOF (Bruker MALDI Biotyper, Bruker Daltonics, Bremen, Germany). In each sample, all morphologically distinct colonies were selected for identification, resulting in multiple isolates per sample and per dog. The characteristics of the isolates are shown in Table 1.

### 4.2. Molecular Analysis

For whole genome sequencing, DNA was isolated using the Qiagen DNA isolation kit (Qiagen, Venlo, The Netherlands). DNA libraries were prepared with the Illumina Nextera kit according to manufacturer’s instructions and sequenced with NextGen paired-end sequencing with 150 bp reads (Illumina, San Diego, CA, USA). Genomes were assembled with SPAdes v3.10.1 [17] and annotated using Prokka v1.13 [18]. Resistance genes were determined using Resfinder [19] and Multi Locus Sequence Type (MLST) was determined with MLSTFinder [20]. Core–gene alignment was performed using Parsnp v1.2 [21]. Gubbins was used to filter recombination regions [22]. SNPs were extracted from the core–gene alignment using SNP-sites v2.4.0 [23] and a minimum-spanning tree (MST) was constructed using the goeBURST algorithm and visualized using Phyloviz v2.0 [24]. Plasmid content was determined using RFPlasmid with a minimum plasmid prediction cut-off of 0.6 and a minimum length of 1kb [25]. The plasmid contigs were characterized using BLASTn.

Isolates containing the combination of resistance genes *ant(6)-Ia*; *aph(3’)-III*; *cat(pC221);* and *erm(B)* were analyzed for the presence of the pRE25-like element. This element has been previously described in *S. pseudintermedius* and was identified using Geneious version 2020.1.1 (Biomatters, Auckland, New Zealand).

### 4.3. Data Availability

Whole genome sequence reads of the canine isolates have been uploaded in ENA under bio project PRJEB53745 and the human isolates were previously uploaded under the following accession numbers: 17S01534-1 (GCA_903992455.1), 16S06119-2 (GCA_903991985.1).

## Figures and Tables

**Figure 1 pathogens-11-00850-f001:**
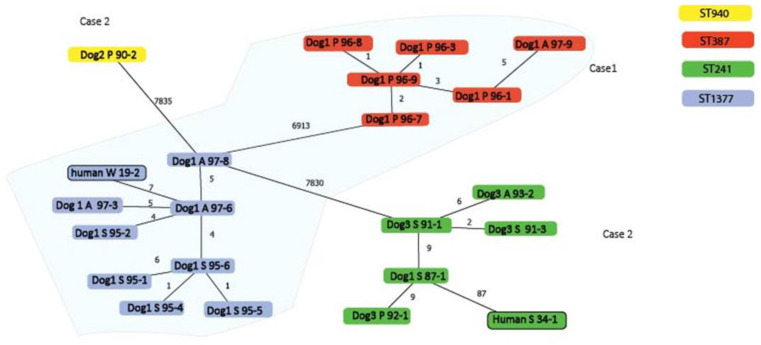
Minimum spanning tree of core-genomes showing the phylogenetic relationship between isolates from the two households, with the number of SNPs indicated on the branches. Isolates are identified by host species, followed by isolation site A = axillary, P = perineum, S = skin, W = wound, and lastly followed by the last three digits of their isolate number. Isolates from household 1 are shown against a blue background. Isolates from household 2 are shown against a white background. Isolates with no SNP differences are not shown.

**Table 1 pathogens-11-00850-t001:** Isolate characteristics.

Isolate	Origin	Isolation Date	Specimen	MLST	Resistance Genes					Mobile Elements
**Household 1**										
16S06119-2	human	June 2016	wound	1377	*blaZ*	*tet*(M)				
16S06095-1	dog 1	June 2016	skin	1377	*blaZ*	*tet*(M)				
16S06095-2	dog 1	June 2016	skin	1377	*blaZ*	*tet*(M)				
16S06095-4	dog 1	June 2016	skin	1377	*blaZ*	*tet*(M)				
16S06095-5	dog 1	June 2016	skin	1377		*tet*(M)				
16S06095-6	dog 1	June 2016	skin	1377	*blaZ*	*tet*(M)				
16S06097-3	dog 1	June 2016	axillary	1377	*blaZ*	*tet*(M)				
16S06097-6	dog 1	June 2016	axillary	1377	*blaZ*	*tet*(M)				
16S06097-7	dog 1	June 2016	axillary	1377	*blaZ*	*tet*(M)				
16S06097-8	dog 1	June 2016	axillary	1377	*blaZ*	*tet*(M)				
16S06096-1	dog 1	June 2016	perineum	387	*blaZ*					
16S06096-3	dog 1	June 2016	perineum	387	*blaZ*					
16S06096-7	dog 1	June 2016	perineum	387	*blaZ*					
16S06096-8	dog 1	June 2016	perineum	387	*blaZ*					
16S06096-9	dog 1	June 2016	perineum	387	*blaZ*					
16S06097-9	dog 1	June 2016	axillary	387	*blaZ*					
**Household 2**										
17S01534-1	human	July 2017	skin	241	*blaZ*	*sat4*	*cat_(pC221)_*	*erm*(B)	*ant(6)-Ia,aph(3’)-III*	PRE25-like; p222
17S01587-1	dog 1	July 2017	skin	241	*blaZ*	*sat4*	*cat_(pC221)_*	*erm*(B)	*ant(6)-Ia,aph(3’)-III*	PRE25-like; p222; 2,7 kb plasmid
17S01591-1	dog 3	July 2017	skin	241	*blaZ*	*sat4*	*cat_(pC221)_*	*erm*(B)	*ant(6)-Ia,aph(3’)-III*	PRE25-like; p222; 2,7 kb plasmid
17S01591-2	dog 3	July 2017	skin	241	*blaZ*	*sat4*	*cat_(pC221)_*	*erm*(B)	*ant(6)-Ia,aph(3’)-III*	PRE25-like; p222; 2,7 kb plasmid
17S01591-3	dog 3	July 2017	skin	241	*blaZ*	*sat4*	*cat_(pC221)_*	*erm*(B)	*ant(6)-Ia,aph(3’)-III*	PRE25-like; p222; 2,7 kb plasmid
17S01592-1	dog 3	July 2017	perineum	241	*blaZ*	*sat4*	*cat_(pC221)_*	*erm*(B)	*ant(6)-Ia,aph(3’)-III*	PRE25-like; p222; 2,7 kb plasmid
17S01593-2	dog3	July 2017	axillary	241	*blaZ*	*sat4*	*cat_(pC221)_*	*erm*(B)	*ant(6)-Ia,aph(3’)-III*	PRE25-like; p222; 2,7 kb plasmid
17S01590-2	dog 2	July 2017	perineum	940	*blaZ*	*sat4*	*cat_(pC221)_*	*erm*(B)	*ant(6)-Ia,aph(3’)-III*	PRE25-like; p222

## Data Availability

Whole genome sequence reads of the canine isolates have been uploaded in ENA under the bio project PRJEB53745 and of the human isolates had previously been uploaded under the following accession numbers: 17S01534-1 (GCA_903992455.1), 16S06119-2 (GCA_903991985.1).

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
