# Peer review of "Within-Household Transmission and Bacterial Diversity of Staphylococcus pseudintermedius"

_pathogens, 2022, doi:10.3390/pathogens11080850_

Round 1

Reviewer 1 Report

In the presented case report the authors describe two cases of human infections caused by the animal-associated Staphylococcus pseudintermedius, regularly isolated from dogs. For this reason, they analyzed the dogs of the infected patients and isolated multiple S. pseudintermedius clones. Whole-genome sequencing revealed that the dog from patient 1 was colonized with two different strains of S. pseudinermedius, of which one was also the causative agent of the human infection. The second patient had three dogs in her household. The isolated strains belonged all to the same sequence type (ST), except one single isolate which belonged to an unrelated ST. Although the patient’s isolate exhibited more single nucleotide polymorphisms (SNPs) than the dogs isolates, the zoonotic origin was quite obvious. The higher number of SNPs could indicate that the transmission from dog to human occurred already a while ago and there was some evolution on the human host. This could also be reflected by the loss of a plasmid which was present in all the related dog isolates but not in the patient. The manuscript is well written and drawn conclusions are valid and available literature has been properly cited.

Minor comments:

Fig. 1: it would be helpful to include the ST numbering also in the figure (corresponding to the color code)

Line 90: it should be “dog 2” instead of “dog 3”

Line 123: “ST940” instead of “ST490”

Line 143: there should be no “,” after “sampled”

The discussion could be improved with the cross reference on Staphylococcus aureus, where host-jumps have been frequently reported, often associated with the loss or gain of specific gene functions or entire plasmids like in the presented case study. (for example https://pubmed.ncbi.nlm.nih.gov/35762208/ or https://pubmed.ncbi.nlm.nih.gov/31124433/)

Author Response

Manuscript Title:

Within-household transmission and bacterial diversity of Staphylococcus pseudintermedius

Journal: Pathogens

Manuscript ID: pathogens-1819378

Reviewer 1:

In the presented case report the authors describe two cases of human infections caused by the animal-associated Staphylococcus pseudintermedius, regularly isolated from dogs. For this reason, they analyzed the dogs of the infected patients and isolated multiple S. pseudintermedius clones. Whole-genome sequencing revealed that the dog from patient 1 was colonized with two different strains of S. pseudinermedius, of which one was also the causative agent of the human infection. The second patient had three dogs in her household. The isolated strains belonged all to the same sequence type (ST), except one single isolate which belonged to an unrelated ST. Although the patient’s isolate exhibited more single nucleotide polymorphisms (SNPs) than the dogs isolates, the zoonotic origin was quite obvious. The higher number of SNPs could indicate that the transmission from dog to human occurred already a while ago and there was some evolution on the human host. This could also be reflected by the loss of a plasmid which was present in all the related dog isolates but not in the patient. The manuscript is well written and drawn conclusions are valid and available literature has been properly cited.

We thank the reviewer for the constructive comments.

Minor comments:

Fig. 1: it would be helpful to include the ST numbering also in the figure (corresponding to the color code)

The ST numbering have been added to the figure and taken out of the figure legend text.

Line 90: it should be “dog 2” instead of “dog 3”

This has been corrected

Line 123: “ST940” instead of “ST490”

This has been corrected

Line 143: there should be no “,” after “sampled”

This has been corrected

The discussion could be improved with the cross reference on Staphylococcus aureus, where host-jumps have been frequently reported, often associated with the loss or gain of specific gene functions or entire plasmids like in the presented case study. (for example https://pubmed.ncbi.nlm.nih.gov/35762208/ or https://pubmed.ncbi.nlm.nih.gov/31124433/)

We agree with the suggestion to refer to known gene loss and acquisition events in S. aureus and have added this to the discussion: “The plasmid present in dog isolates in household 2 was absent in the ST241 human isolate and shows that gain or loss of a plasmid occurred among genetically highly related isolates.  This is in line with observed gene loss or acquisition events in S. aureus, involved in the host jump of CC398 from livestock to human, and there are other examples of gene acquisitions in S. aureus that have facilitated adaptations to other animal species [15,16]. The mobile elements in S. pseudintermedius carrying multiple resistance genes and potential virulence genes, are important epidemiological markers to monitor, as they can act as reservoir for transmission to humans.

Reviewer 2 Report

The text focuses very well on the results and the used methods are innovative.

The data are interesting and the paper is clear and very well written in English.  

Author Response

Manuscript Title:

Within-household transmission and bacterial diversity of Staphylococcus pseudintermedius

Journal: Pathogens

Manuscript ID: pathogens-1819378

Reviewer 2:

The text focuses very well on the results and the used methods are innovative.

The data are interesting and the paper is clear and very well written in English.  

We thank the reviewer for the very positive comments.

Reviewer 3 Report

In this case report, the authors show data about the “within-host” diversity of Staphylococcus pseudintermedius isolates from dogs. The study was well performed and the results are clearly described. The discussion is balanced and contains all major findings.

I have three minor suggestions:

1. Introduction: There is a recent paper quantifying the carriage rates of S. pseudintermedius among dogs and humans within households. This information could be added to facilitate estimating the likeliness of transmission for the reader: https://pubmed.ncbi.nlm.nih.gov/35456729/

2. Fig1. Understanding of Fig. 1 could be facilitated, if instead of the numbers, the cells contained an individual ID including the information whether the isolate is from a dog or a human (e.g. instead of “16506096-8” for example DOG—A-8)

3. Results: As the authors demonstrated diversity of the isolates from dogs it would be interesting to know, whether the dogs were exposed to selective antibiotic pressure or were treated in a hospital?

Author Response

Manuscript Title:

Within-household transmission and bacterial diversity of Staphylococcus pseudintermedius

Journal: Pathogens

Manuscript ID: pathogens-1819378

Reviewer 3:

In this case report, the authors show data about the “within-host” diversity of Staphylococcus pseudintermedius isolates from dogs. The study was well performed and the results are clearly described. The discussion is balanced and contains all major findings.

We thank the reviewer for the constructive comments and have adapted the manuscript accordingly.

I have three minor suggestions:

  1. Introduction: There is a recent paper quantifying the carriage rates of S. pseudintermedius among dogs and humans within households. This information could be added to facilitate estimating the likeliness of transmission for the reader: https://pubmed.ncbi.nlm.nih.gov/35456729/

The reference was added along with the sentence “Nevertheless, carriage rates of S. pseudintermedius in humans remain very low compared to the carriage rates of dogs, even in dog owning households [6].” line 36. 

  1. Fig1. Understanding of Fig. 1 could be facilitated, if instead of the numbers, the cells contained an individual ID including the information whether the isolate is from a dog or a human (e.g. instead of “16506096-8” for example DOG—A-8)

The figure has been modified and “Isolates are identified by host species, followed by isolation site A = axillary, P = perineum, S = skin, W = wound, followed by the last three digits of their isolate number.” has been added in the figure legend text.

  1. Results: As the authors demonstrated diversity of the isolates from dogs it would be interesting to know, whether the dogs were exposed to selective antibiotic pressure or were treated in a hospital?

We agree with the reviewer that this information would be of great interest, but unfortunately, we could not retrieve any information if the dogs were treated with antibiotics prior to sampling. The owners had sampled the dogs that were residing in the household at the time they had clinical signs of an S. pseudintermedius infection.